# Use of Levosimendan in Patients with Advanced Heart Failure: An Update

**DOI:** 10.3390/jcm11216408

**Published:** 2022-10-29

**Authors:** Daniele Masarone, Michelle M. Kittleson, Piero Pollesello, Marco Marini, Massimo Iacoviello, Fabrizio Oliva, Angelo Caiazzo, Andrea Petraio, Giuseppe Pacileo

**Affiliations:** 1Heart Failure Unit, Department of Cardiology, AORN dei Colli-Monaldi Hospital Naples, 80131 Naples, Italy; 2Department of Cardiology, Smidt Heart Institute, Cedars-Sinai, Los Angeles, CA 90048, USA; 3Critical Care, Orion Pharma, 02101 Espoo, Finland; 4Cardiology Division, Cardiovascular Department, Azienda Ospedaliero Universitaria Ospedali Riuniti di Ancona Umberto I-GM Lancisi-G Salesi, 60126 Ancona, Italy; 5Intensive Cardiac Care Unit, De Gasperis Cardio Center, ASST Grande Ospedale Metropolitano Niguarda, 20162 Milan, Italy; 6Department of Medical and Surgical Sciences, University of Foggia, 71122 Foggia, Italy; 7Heart Transplant Unit, Department of Cardiac Surgery and Transplant, AORN dei Colli-Monaldi Hospital, 80131 Naples, Italy

**Keywords:** advanced heart failure, inodilators, levosimendan, pharmacologic therapy

## Abstract

Levosimendan is an inodilator drug that, given its unique pharmacological actions and safety profile, represents a viable therapeutic option in patients with heart failure with reduced ejection fraction in the advanced stage of the disease (advHFrEF). Pulsed levosimendan infusion in patients with advHFrEF improves symptoms and clinical and hemodynamic status, prevents recurrent hospitalizations, and enables optimization of guidelines-directed medical therapy. Furthermore, considering its proprieties on right ventricular function and pulmonary circulation, levosimendan could be helpful for the prevention and treatment of the right ventricular dysfunction post-implanting a left ventricular assist device. However, to date, evidence on this issue is scarce and has yielded mixed results. Finally, preliminary experiences indicate that treatment with levosimendan at scheduled intervals may serve as a “bridge to transplant” strategy in patients with advHFrEF. In this review, we summarized the clinical pharmacology of levosimendan, the available evidence in the treatment of patients with advHFrEF, as well as a hypothesis for its use in patients with advanced heart failure with preserved ejection fraction.

## 1. Introduction

Although pharmacologic and non-pharmacologic treatment of patients with heart failure (HF) with reduced ejection fraction (HFrEF) improves quality of life and survival rates [1], a variable percentage (up to 13%) of patients do not respond to conventional therapy, resulting in progression to the more advanced stage of the disease (advHFrEF) [2]. Because patients with advHFrEF often have a reduced tolerance to disease-modifying drugs [3], inotropes are frequently used to improve symptoms and quality of life and to reduce hospitalizations [4]. Among the inotropes, levosimendan has been demonstrated to achieve these goals in patients with advHFrEF [5]. In addition, it has been shown that levosimendan may be helpful in the prevention and treatment of the right ventricular dysfunction after a left ventricular assist device (LVAD) implant and as a “bridge to transplant” strategy in patients on a waiting list for a heart transplant [6]. In this review, we summarize the clinical pharmacology of levosimendan and the research outcomes for levosimendan in patients with advHFrEF, while also providing practical advice regarding the use of levosimendan in clinical practice

## 2. Pharmacology of Levosimendan

### 2.1. Pharmacokinetic of Levosimendan

Levosimendan is a pro-drug that presents linear kinetics without renal and hepatic impairment [7]. In clinical practice, it is administered intravenously, but levosimendan has an oral bioavailability of 85%, a volume of distribution of 0.2 L/Kg, and very high plasma protein binding (97–98%) [8]. In addition, levosimendan is extensively metabolized before excretion in urine and feces, primarily through conjugation with glutathione to form inactive metabolites [8]. The minor route of metabolization (approximately 6% of the total dose of levosimendan) is the intestinal transformation in an intermediate metabolite (OR-1855), which is further metabolized by acetylation into the active metabolite (OR-1896, Figure 1). 

Levosimendan has a half-life of around 1 h; therefore, even in patients with HFrEF, it has a rapid elimination from circulation at the end of the infusion. On the other hand, the half-life of levosimendan metabolites is roughly 80 h, with both OR-1855 and OR-1896 reaching their peak plasma concentrations at 48–72 after the levosimendan administration. This means that the pharmacodynamic effects persist for 10–14 days after infusion [9,10]. Comparative studies have shown that levosimendan’s pharmacokinetics are not significantly affected by HF, mild to moderate renal and hepatic impairment [11,12]. In contrast, in patients with severe renal dysfunction, the half-life of OR-1855 and OR-1896 is prolonged by 1.5, and their area under the curve and peak concentrations are 2-fold higher [13].

Therefore, it has been suggested that the dose and infusion rate be reduced when levosimendan is used in patients with severe chronic kidney disease [14].

### 2.2. Pharmacodynamics of Levosimendan

Levosimendan possesses a triple mechanism of action [15]. First, the inotropic effect is due to calcium sensitization achieved through selective binding to the calcium-bound form of cardiac troponin C, resulting in increased cardiac contractility in the absence of alterations in cardiomyocyte electrophysiological homeostasis and with myocardial relaxation [16]. The second mechanism is the activation (resulting in the opening) of K^+^ adenosine triphosphate (ATP)-dependent channels present in vascular smooth muscle cells; this mechanism results in improved oxygen delivery to the myocardium in the absence of increased oxygen demand [17] while also promoting arterial and venous vasodilation [18,19]. The third mechanism is the opening up of ATP-dependent K^+^ channels in the mitochondria, producing a cardioprotective and organ-protective effect [20,21]. Finally, systemic effects have also been demonstrated, including anti-inflammatory [22] and antiapoptotic effects [23], although the clinical relevance of these effects is uncertain.

### 2.3. Side Effects and Contraindications of Levosimendan

Levosimendan is generally well tolerated in patients with HF. The most common adverse effects are secondary to vasodilatation and include hypotension, headache, and nausea [24].

Regarding arrhythmias, levosimendan infusion is associated with an increased incidence of atrial fibrillation compared with dobutamine and placebo [25,26].

However, unlike the other inotropes, levosimendan does not increase intracellular calcium concentration and myocardial oxygen consumption, meaning that ventricular arrhythmias are unlikely during levosimendan treatment [27].

Finally, hypokalemia is a typical side effect of levosimendan administration, but the mechanism responsible for this effect is not yet known [28].

Contraindications to the use of levosimendan include severe symptomatic hypotension (systolic blood pressure <70 mmHg), significant mechanical obstruction affecting ventricular filling or outflow, or both (i.e., severe mitral stenosis, severe aortic stenosis), severe renal impairment (i.e., creatinine clearance <30 mL/min/1.73 m^2^), and severe hepatic impairment (i.e., MELD score >30).

## 3. Intermittent Levosimendan Infusion in Patients with advHFrEF

Several small-scale, non-randomized trials and registries of advHFrEF patients not eligible or waiting for a heart transplant or an LVAD implant have shown that repeated infusions of levosimendan improve symptoms [29] and clinical and hemodynamic status [30,31,32], prevent recurrent hospitalizations [33], and enable the optimization of guideline-directed medical therapy [34]. However, as highlighted in Table 1, different administration protocols were used in these studies, so the optimal administration strategy has not yet been identified.

Three randomized clinical trials on the use of levosimendan in advHFrEF patients have also been conducted. The trial LevoRep (efficacy and safety of the pulsed infusions of levosimendan in outpatients with advanced heart failure) enrolled 120 outpatients with advHFrEF randomized to levosimendan (0.2 µg/kg/min for 6 hours at 2-week intervals over 6 weeks) or placebo [40]. In this trial, levosimendan failed to achieve the primary endpoint (a composite endpoint of improvement in the 6-min walk test ≥20% and increase in score on the Kansas City Cardiomyopathy Questionnaire ≥15%) (19% vs. 15%; OR 1.25; 95% CI 0.44–3.59; *p* = 0.810). In the LIONHEART (efficacy and safety of intermittent intravenous outpatient administration of levosimendan in patients with advanced heart failure) trial, 69 patients with advHFrEF were randomized to levosimendan (0.2 μg/kg/min for 6 hours every 2 weeks for 12 weeks) versus placebo [41]. At the end of the study, patients in the levosimendan arm significantly reduced NT-proBNP plasma levels more than the placebo group (mean change in NT-proBNP–1446 vs.–1320 pg/mL; *p* < 0.001). Moreover, the patients treated with levosimendan experienced a reduction in HF-related hospitalization (HR 0.25; 95% CI 0.11–0.56; *p* = 0.001) and were shown to have the lowest probability of a clinically significant decline in quality of life (*p* = 0.022).

In the LAICA (efficacy and safety of intermittent repeated levosimendan infusions in advanced heart failure patients) study, 97 patients were randomized to levosimendan (0.1 μg/kg/min as a continuous 24-h intravenous infusion administered once monthly for 1 year) vs. placebo [42]. In this trial, levosimendan did not reduce the rate of readmissions for acute decompensated HF (HR 0.66; 95% CI, 0.32–1.32; *p* = 0.24). However, patients in the treatment arm exhibited a significantly lower cumulative incidence of acute decompensation of HF and/or death at 1 month (5.7% vs. 25.9%; *p* = 0.004) and 3 months (17.1% vs. 48.1%; *p* = 0.001) and a significant improvement in survival during 12 months of treatment (log-rank: 4.06; *p* = 0.044).

A recent meta-analysis of 984 patients (727 treated with levosimendan and 257 in the control group) showed that levosimendan treatment was associated with an improvement in NYHA class (*p* < 0.001), left ventricular ejection fraction (*p* < 0.001), as well as a reduction in natriuretic peptide levels (*p* < 0001) [43]. Furthermore, although all-cause mortality did not differ between the two groups, cardiovascular death was lower in levosimendan-treated patients than in controls (*p* = 0.02). Taking into account the data from both these studies and the meta-analyses [44,45], the generally accepted conclusion is that the repetitive application of levosimendan is likely to be effective, feasible, and safe in patients with advHFrEF. Furthermore, the author believes that both 6-h and 24-h pulsed administration of levosimendan are effective in patients with advHFrEF.

The ongoing trial LEODOR (Repetitive Levosimendan Infusion for Patients with Advanced Chronic Heart Failure trial; NCT03437226) will test the efficacy and safety of intermittent levosimendan therapy in patients with advHFrEF in the vulnerable phase, offering additional evidence regarding the use of levosimendan in this challenging patient population [46]. The LEIA-HF (Levosimendan In Ambulatory Heart Failure Patients; NCT04705337) is another multicenter, randomized, double-blind, placebo-controlled trial whose purpose is to evaluate whether the repetitive use of continuous 24-h infusions of levosimendan every 4 weeks for 48 weeks reduces the incidence of adverse cardiovascular events in outpatients with chronic advHFrEF [47].

## 4. Levosimendan in Patients with advHFrEF Undergoing LVAD Implantation

Left ventricular assist device (LVAD) implant is an effective management strategy for patients with advHFrEF [48]. In particular, the new generation devices (specifically the HeartMate 3) have short- and medium-term survival rates comparable to heart transplantation [49]. Unfortunately, though, up to 25% of patients who undergo LVAD implantation develop post-implanted right ventricular heart failure with significantly increased morbidity and mortality rates [50]. Given the pharmacological effects of levosimendan on the right ventricle and pulmonary circulation [51,52], the impact of pretreatment with levosimendan on right ventricular dysfunction after LVAD implantation has been evaluated.

Sponga et al. analyzed, in a single-center study, the effects of levosimendan infusion on hemodynamic parameters in patients with borderline right ventricular function before urgent LVAD implantation and the prognostic effect of response to levosimendan infusion [53]. Treatment with levosimendan resulted in a dose-dependent increase in cardiac index by 21% (*p* = 0.014), a decrease in pulmonary pressure by 12% (*p* = 0.003), S and a decrease in pulmonary capillary wedge pressure and central venous pressure by 15% (*p* = 0.028 and *p* = 0.016). Notably, hemodynamic improvements persisted for 24 h after discontinuing levosimendan infusion in patients who survived but not in those who subsequently died of right ventricular failure. Based on these results, the authors stated that hemodynamic response after levosimendan infusion could predict mortality and right ventricular dysfunction in advHFrEF patients undergoing urgent LVAD implantation. In a retrospective post hoc analysis, 9 patients with LVAD support received levosimendan without experiencing any adverse effects. At 24 months, the survival rate was 89%, which is a better result than that seen in the data from the fifth INTERMACS registry, which reports a 2-year survival of 75% [54]. However, the lack of a control group does not allow firm conclusions to be made regarding the benefit of levosimendan in these patients; also, in this study, post-LVAD right ventricular dysfunction was not assessed.

In a retrospective single-center study, 85 patients with advHFrEF and LVAD exhibited improved right ventricular stroke work index (406.26 ± 251.30 vs. 275.48 ± 200.51 g/m^2^/b/min; *p* = 0.025) and reduced pulmonary vascular resistance (4.0 ± 1.8 vs. 3.0 ± 1.4 wood units; *p* = 0.038) when levosimendan was added to other inotropes; however, no significant difference in early and late right ventricular dysfunction occurred [55]. In another single-center study, 84 patients with advHFrEF who underwent LVAD implant were randomized to levosimendan and placebo. No difference in the right ventricular failure rate was observed between the two groups (7.5% vs. 13.6%; *p* = 0.43) as well as no significant difference in in-hospital (5% vs. 4.5%; *p* > 0.999) and long-term mortality (10% vs. 27.3%; *p* = 0.64) rates [56].

Moreover, a recent meta-analysis of 106 patients with advHFrEF who underwent LVAD implant [57] showed that levosimendan administration was associated with hemodynamic improvements and improved organ perfusion. However, such hemodynamic benefits are not associated with a reduction in mortality, which is likely a result of the low statistical power of the studies conducted to date.

A multicenter randomized, placebo-controlled trial is needed to obtain conclusive results.

## 5. Levosimendan in Patients with advHFrEF on the Waiting List for a Heart Transplant

Heart transplantation remains the gold-standard treatment for selected patients with advanced HF [58].

However, organ shortages continue to limit the number of transplants that can be performed each year, thus increasing the waiting time for patients to receive a compatible and suitable heart [59].

Intermittent use of levosimendan may be helpful in this challenging clinical setting. For example, in a single-center study, 11 patients on the waiting list for heart transplantation [60] were given scheduled infusions of levosimendan (a 6-h infusion every 2 months at a dose of 0.1–0.2 mg/kg/min, depending on the patient’s blood pressure). This therapeutic strategy reduced both the rate of rehospitalization and the need for urgent heart transplantation (22% vs. 44% in Spanish registries). Although these results are preliminary and inconclusive, expert consensus points to levosimendan as a viable therapeutic option as a bridge to transplantation [61] in patients who are not candidates for LVAD to ensure adequate end-organ perfusion (and thus prevent the onset of multiorgan failure) and to avoid increased pulmonary vascular pressures and resistances (and therefore avoid patient exclusion from the heart transplant waiting list or the need for heart-lung transplantation).

## 6. Levosimendan in Patients with advHFpEF as a Future Perspective

In this review, we summarize the available evidence on the use of levosimendan in patients with advHFrEF; however, a sizeable proportion of patients with advanced HF have a preserved ejection fraction (advHFpEF) [62]. In addition, such patients have unique hemodynamic features such as a persistent elevation of pulmonary capillary wedge pressures (PCWP) and pulmonary pressure at rest or during exertion as well as an inability to appropriately augment the cardiac index during exercise [63,64].

In the Levosimendan Improves Hemodynamics and Exercise Tolerance in PH-HFpEF (HELP) trial, 37 patients with advHFpEF were randomized to levosimendan and placebo. In this preliminary study, levosimendan reduced PCWP during exercise (−3.9 ± 2.0 mm Hg; *p* = 0.047) with a trend in the increase of cardiac index during exercise (2.5 ± 0.8 at baseline vs. 3.2 ± 1.1 at 25 watts). Furthermore, levosimendan treatment resulted in a 29.3-meter rise in the distance walked during the 6-minute walking test compared with placebo (95% CI: 2.5 to 56.1; *p* = 0.033) [65].

Although these data are preliminary, further studies may confirm that levosimendan improves exercise capacity and quality of life in patients with advHFpEF.

## 7. Tips and Tricks for the Use of Levosimendan in Clinical Practice

In the previous sections, we reviewed the evidence on the use of levosimendan in patients with advanced heart failure; in this section, we will offer practical advice regarding how to use levosimendan in patients with advHFrEF to facilitate the use of this drug in common clinical practice.

According to European Society of Cardiology guidelines [66], periodic infusion of levosimendan may be considered a palliative strategy or as a “bridge to transplant/LVAD” strategy in patients with advHFrEF with evidence of organ hypoperfusion.

For both indications, we recommend the first administration of levosimendan be performed in an inpatient setting and in 24 h at a dose of 0.2 µg/kg/min to verify both safety (particularly in terms of the appearance of symptomatic hypotension and ventricular tachycardias) and efficacy. For palliative purposes, the response to levosimendan infusion can be assessed as a subjective reduction of symptoms and improvement of quality of life; in doubtful cases, the assessment of NT-proBNP plasma values can be helpful. In contrast, in the case of a “bridge to transplant/LVAD” strategy, we recommend objectifying the efficacy of levosimendan by echocardiography (improvement of biventricular systolic function, reduction of pulmonary circulation pressures) or, in doubtful cases, by right heart catheterization.

Subsequent dosing can be given in either 24-h or 6-h periods, depending on the patient’s profile (Figure 2). 

In patients with systolic blood pressure >100 mmHg, mildly or moderately reduced renal function (estimated glomerular filtrate > 45 mL/min/1.73 m^2^), and no history of complex ventricular arrhythmias, we perform administration of 6.25 mg levosimendan at a dosage of 0.2 µg/kg/min every two weeks.

In contrast, in patients with systolic blood pressure <100 mmHg, severely reduced renal function (estimated glomerular filtrate >30 <45 mL/min/1.73 m^2^), and a history of complex ventricular arrhythmias, we recommend administration of 12.5 mg levosimendan at a dosage of 0.1 µg/kg/min every four weeks.

The latter administration scheme can also be used in carefully selected advHFrEF patients with glomerular filtrate >15 mL/min/1.73 m^2^ <30 mL/min/1.73 m^2^ when levosimendan infusion for palliative purposes documents marked improvement in symptoms and quality of life.

Finally, in patients with an indication for LVAD implantation, we recommend the day before the implant, 24-h administration of levosimendan at a dose of 0.2 µg/kg/min combined with noradrenaline or adrenaline 0.1–0.2 µg/kg/min in patients with a high risk of right ventricular dysfunction post-implantation of LVAD (e.g., patients with right ventricular failure risk score >5.5).

## 8. Conclusions

With its unique pharmacological action and safety profile, levosimendan represents a viable therapeutic option in patients with advHFrEF to prevent HF-related hospitalizations, improve quality of life, and serve as a “bridge to transplant” strategy. In its first twenty years, levosimendan has been transformed from an innovative infusion for the management of acute HF to a safe and potentially effective option for outpatients with advHFrEF. 

Over the next several years, randomized trials will hopefully establish a role for levosimendan in preventing right ventricular dysfunction post LVAD implantation and in the treatment of advHFpEF.

## Figures and Tables

**Figure 1 jcm-11-06408-f001:**
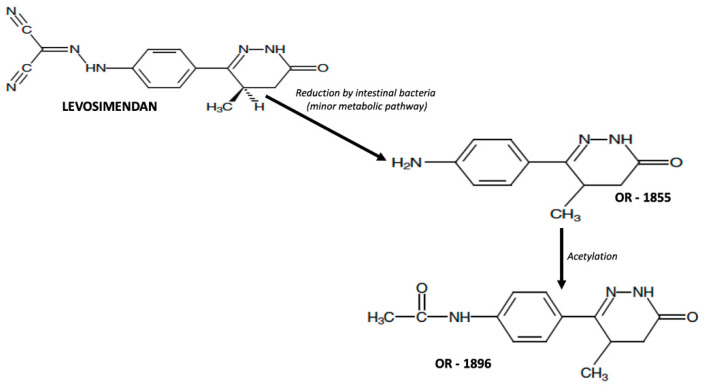
Metabolic pathway of transformation of levosimendan in its active metabolites.

**Figure 2 jcm-11-06408-f002:**
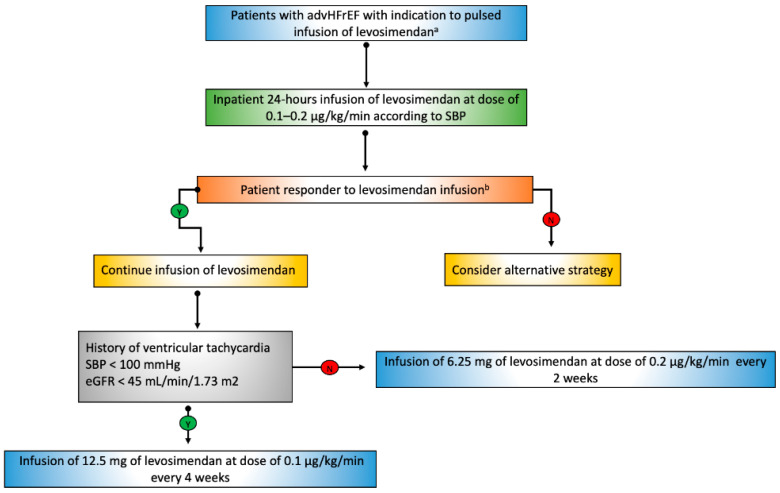
Therapeutic algorithm for the use of pulsed infusion of levosimendan in patients with advHFrEF. SBP: systolic blood pressure, eGFR: estimated glomerular filtration rate. a: Indication to pulsed infusion of levosimendan INTERMACS Class IV (frequent-flyers patients), progressive deterioration of kidney function, combined precapillary and post-capillary pulmonary hypertension, persistently high levels of NT-proBNP despite guidelines-directed medical therapy. b: subjective reduction of symptoms and improvement of quality of life; in doubtful cases, the assessment of NT-proBNP plasma values (palliative purpose); improvement of biventricular systolic function and reduction of pulmonary circulation pressures.

**Table 1 jcm-11-06408-t001:** Summary of the clinical study on the repetitive infusion of levosimendan in patients with advHFrEF.

Study	N° of Patients	Levosimendan Dose	Time of Infusion	Interval of Infusion	Results
Nanas 2005 [35]	36	Bolus dose (6 mg/kg) plus Infusion rate (0.2 mcg/Kg/min). Levosimendan was added to dobutamine infusion	24 h	2 weeks for 45 days	Improvement in survival (6% vs. 61% *p* = 0.0002)
Parissis 2006 [36]	25	Bolus dose (6 mg/kg) plus Infusion rate (0.1–0.4 mcg/Kg/min)	24 h	3 weeks for 114 days	Reduction of LVEDVi (120 vs. 156 mL/m^2^; *p* < 0.01), LVESVi (80 vs. 106 mL/m^2^; *p* < 0.01) and NT-proBNP plasma levels (966 vs. 1529 pg/mL; *p* < 0.01) increase of LVEF (26 vs. 22%, *p* < 0.01)
Mavrogeni 2007 [37]	50	Bolus dose (6 mg/kg) plus Infusion rate (0.1–0.2 mcg/Kg/min)	24 h	30 days for 6 months	Increase of LVEF (28 + 7 vs. 21 + 4%, *p* = 0.003) and LVFS (15 + vs. 11 + 3%, *p* = 0.006).
Papadopoulou [38]	20	No bolus dose Infusion rate (0.1 mcg/kg/min)	24 h	30 days for 6 months	Increase of LVEF (30.3 ± 6.9 vs. 32.1 ± 7.4%; *p* = 0.01) and quality of life (LIhFE score i 35.4 ± 18.6 vs. 22.2 ± 13.0; *p* < 0.0001).
Malfatto 2012 [39]	33	No bolus dose Infusion rate (0.1–0.4 mcg/kg/min)	24 h	30 days for 12 months	Increase of LVEF (25.9 + 5.1 vs. 28.7 ± 5.4%; *p* < 0.05) and CI (2.34 + 0.58 vs. 2.77 + 0.65 L/min/m2; *p* < 0.05). Reduction of PASP (51.8 ± 15.4 vs. 42.6 ± 13.0 mmHg; *p* < 0.05), E/e’ ratio (18.3 ± 8.9 vs. 13.8 ± 4.1; *p* < 0.05)
Oliva (RELEVANT-HF) 2018 [33]	185	No bolus dose Infusion rate (0.2 mcg/Kg/min)	24 h	3–4 weeks for 6 months	Reduction of days in hospital (9.4 vs. 2.8 days; *p* < 0.0001) and length of HF admissions (17.4 vs. 21.6 days; *p* = 0.0001)
Masarone 2020 [29]	15	No bolus dose Infusion rate (0.2 mcg/Kg/min	6 h	2 weeks for 12 months	Reduction of HF-related hospitalizations (2 vs. 10; *p* < 0.05) and increase of distance walked at six-minute walking test (282 ± 52 vs. 248 ± 30 meters; *p* < 0.05)
Altenberger (LevoRep) 2014 [40]	120	No bolus dose Infusion rate (0.2 mcg/Kg/min)	6 h	2 weeks for 42 days	No increase in the distance walked on the 6-minute walking test and no increase in score on the Kansas City Cardiomyopathy Questionnaire (19% vs. 15%; OR.25; 95% CI 0.44–3.59; *p* = 0.810).
Comín-Colet (LION HEART) 2018 [41]	69	No bolus dose Infusion rate (0.2 mcg/Kg/min)	6 h	2 weeks for 6 months	Reduction of NT-proBNP plasma levels (mean change in NT-proBNP–1446 vs. –1320 pg/mL; *p* < 0.001) and of the rate of HF-related hospitalization (hazard ratio 0.25; 95% CI 0.11–0.56; *p* = 0.001)
García-González (LAICA) 2021 [42]	97	No bolus dose Infusion rate (0.1 mcg/Kg/min)	24 h	4 weeks for 12 months	No reduction in HF-related hospitalizations (HR 0.66; 95% CI, 0.32–1.32; *p* = 0.24). Reduction of cumulative incidence of HF-related hospitalizations and death at 1 month (5.7% vs. 25.9%; *p* = 0.004) and 3 months (17.1% vs. 48.1%; *p* = 0.001). Improvement in survival (log-rank: 4.06; *p* = 0.044).

LVEDVi: left ventricular end-diastolic diameter index, LVESVi: left ventricular end-systolic volume index, NT-proBNP: N-terminal pro-brain natriuretic peptides, LVEF: left ventricular ejection fraction, LVFS: left ventricular fractional shortening, CI: cardiac index, PASP: pulmonary artery systolic pressure, HF: heart failure.

## Data Availability

Not available.

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
