# Peer review of "Use of Levosimendan in Patients with Advanced Heart Failure: An Update"

_jcm, 2022, doi:10.3390/jcm11216408_

Round 1

Reviewer 1 Report

 Masarone and colleagues provides an update about the use of Levosimendan in patients with advanced heart failure.

I have the following comments:

1.       Major concern regarding this paper is the of clarity regarding the novelty aspect in the data summarized compared to most recent literature. Interestingly, the author neglected important literature particularly the following three papers:

a)       In the section about (Intermittent levosimendan infusion in patients with advHF), the authors did not refer to an important systematic review and meta-analysis(1). Could you cite and include this important systematic review in your review?

b)       In the section about (Levosimendan in patients with advHFrEF undergoing LVAD implantation), the authors did not refer to an important systematic review(2). Could you cite and include this important systematic review in your review?

c)       The authors did not mention the use of Levosimendan for the treatment of a late RHF patient post-LVAD(3). Please add this to the section about (Levosimendan in patients with advHFrEF undergoing LVAD implantation).

2.       The authors should elaborate more on the safety of Levosimendan and the incidence of arrhythmia, hypotension, and vasoplegia.

3.       The authors should add their conclusion and experience about the preferred Levosimendan infusion protocol in different HF indications.

4.       The section about (Future perspective), the title does not reflect the text. Please change the title to (Levosimendan in patients with advHFpEF as a future perspective).

5.       The authors should add comments regarding the availability of Levosimendan in the USA and the lack of FDA approval. May be to be added a section about the history of Levosimendan and its current registration status.

6.       The numbers of the last three sections are wrong (repeated number 5) please review and correct these numbers.  

  References

1.           Elsherbini H, Soliman O, Zijderhand C et al. Intermittent levosimendan infusion in ambulatory patients with end-stage heart failure: a systematic review and meta-analysis of 984 patients. Heart Failure Reviews 2022;27:493-505.

2.           Abdelshafy M, Elsherbini H, Elkoumy A et al. Perioperative Levosimendan Infusion in Patients With End-Stage Heart Failure Undergoing Left Ventricular Assist Device Implantation. Frontiers in cardiovascular medicine 2022;9:888136-888136.

3.           Yalcin YC, Caliskan K. Intermittent levosimendan treatment for late onset right ventricular failure in a patient supported with a left ventricular assist device. Artif Organs 2020;44:533-534.

Reviewer 2 Report

This is a nice narrative review of Pharmacology of Levosimendan and its use in patients with Advanced heart failure undergoing both LVAD insertion or heart transplantation. There are a few comments that needs author’s attention.

In page 2 line 63 "On the 63 other hand, the half-life of OR-1855 and OR-1896 is about 80 hours, ensuring the persistence of pharmacodynamic effects for 10-14 days after drug infusion " If half-life is 80 hours how can the effects last up to 14 days, this sentence needs further explanation.

In table 1 the following study does not have any reference "Scheduled intermittent in-otropes for Ambulatory Advanced Heart Failure. The RELEVANT-HF multi-centre collaboration”

In page 2 line 69 the impact of renal impairment is described, however there is no mention of   drug modifications on patients with advanced liver disease. Please expand the text on this point.

Authors should also reflect on side effects of drug and contraindications of its administration- active infection, sepsis, obstructive hypertrophic cardiomyopathy, Aortic stenosis.

LVAD and heart transplantation setting are discussed but it would be interesting for reader to know evidence of benefit of Levosimendan in patients with HFrEF undergoing cardiac surgery (either valves or CABG)

Introduction and abstract are too similar. Please modify introduction accordingly

Reviewer 3 Report

Masarone et al provide a very nice and succinct review of the literature surrounding the use of levosimendan in advHFrEF. It is well organized and comprehensive. 

My only suggestion is perhaps a section summarizing some of the adverse effects/events attributed to levosimendan use found in these trials which would be a useful guide for clinicians who may be using this medication in the future.